# The Capitalization of School Quality in Rents in the Beijing Housing Market: A Propensity Score Matching Method

Zisheng Song 

Department of Real Estate and Construction Management, Royal Institute of Technology (KTH), 100 44 Stockholm, Sweden; zisheng@kth.se

**Abstract:** In China, the capitalization of education resources in housing prices has been widely discussed. However, insufficient attention is paid to it in rents. Thus, this paper mainly aims to identify the capitalization of school quality in rents. It estimates a hedonic treatment effects model by introducing the propensity score matching (PSM) method. The empirical analysis is based on 49,438 rental transaction data of 2016–2018 in Beijing, China. It finds that school quality can be significantly capitalized in rents across different school quality (ranked as 1st-class, 2nd-class, and popular-class), space, and time. Besides, quality school density (the number of quality schools) within neighborhoods can significantly moderate the nearest school's capitalization, promoting a 3.5% capitalization increase in outer municipal districts but a 3% decrease in inner districts. The popular-class schools can be capitalized into the rent of inner districts, probably because of other exogenous factors (e.g., housing prices, public transit). In addition, the equitable housing policy might show a potential risk in worsening social inequality between homeowners and renters in the municipal areas with high competition for 1st-class schools. In contrast, it may remedy such inequality in outer districts with less competition for quality schools.

**Keywords:** rent; school capitalization; propensity score matching (PSM) method; neighborhood school density; moderating effects; spatial–temporal heterogeneity

**JEL Classification:** C31; C51; R12; R23; R32; R38



## 1. Introduction

Internationally, the issue of school quality capitalization in property values has been analyzed extensively. In countries where school choice is linked to a residential address, it is crucial to move to the "right" school district to access quality schools. There is a large body of international literature on this topic [1–3]. In China, access to school enrollment as a type of property right is highly linked to homeownership and a residential address rather than a rental lease [4]. In other words, rental households might be excluded from such property rights. Since 2016, a series of statements proposed to enable equal rights for tenants to public services as homeowners (e.g., proximity-based primary schools). However, rental households still cannot obtain quality school enrollment as equally as homeowners in megacities, e.g., Beijing and Shanghai. These megacities are characterized by unaffordable housing prices, high population aggregation, and limited public education. Therefore, it is necessary to quantitatively evaluate quality schools' capitalization in rents and consider its effects on social inequality between homeowners and renters.

Many studies have analyzed how school quality is capitalized in housing prices; however, very few studies pay attention to the capitalization of school quality in rents. Existing literature compares the capitalization of quality schools in rents to housing prices in China at the neighborhood level [5]. Zhang and Chen (2018) investigate the rent-yields gap between apartments near different quality schools [6]. They find that unequal enrollment policy can significantly induce severe educational and residential segregation between

homeowners and tenants. Further, some recent studies focus on evaluating the equitable housing policy, which is implemented at the end of 2017 and aims to grant tenants the equal right to access the neighborhood primary schools as homeowners. For example, Hu et al. (2020) find that school quality can significantly be capitalized in rents and worsen social inequality between homeowners and renters [7]. However, these studies rarely discuss school capitalization across high-ranked quality schools, spatial heterogeneity, and moderating effects of quality school density in neighborhoods. Therefore, this paper aims to construct a treatment-effect hedonic model and estimate quality school capitalization in rents from a space–temporal perspective.

This paper might provide several contributions. First, it constructs a treatment-effect hedonic model and regards the quality school as a treatment for rental housing by using the propensity score matching (PSM) method. Second, this paper extensively discusses the heterogeneity of school capitalization in rents across different school quality rankings. This discussion would extend the existing literature that does not discuss in-depth schools of different quality ranking can be capitalized differently in rents. Third, this paper further detects the moderating effect of school density in neighborhoods on school quality capitalization. School density can amplify the third ranked schools' capitalization in suburban districts and weaken the first ranked schools' capitalization in inner districts of high competition for quality schools. Fourth, it further explores the heterogeneity of school capitalization in different residential zones and periods before and after the equitable right policy.

The rest of this paper is organized as follows: Section 2 provides a literature review on school capitalization, casual effects measurements, and studies of school capitalization in the context of China. Section 3 gives an introduction to the case of Beijing. Section 4 presents the data and models in this project. The findings and discussions of this paper are in Section 5. Section 6 provides the conclusions.

## 2. Literature Review

This section presents a brief review of the literature on the capitalization of quality schools and related studies conducted in China. Additionally, this section also gives a review of school capitalization measurements and treatment effects analyses.

### 2.1. School Quality Capitalization

Many studies have found a strong relationship between school quality and housing values, arguing that quality schools have a positive impact on housing prices [7–13]. Oates (1969) initiated the study of school quality capitalization and argued that housing prices incorporated the capitalization of education quality and local property taxes [8]. Brasington (2001, 2002, and 2003) extended Oates's argument in a handful of studies. He claimed that housing prices were more sensitive to changes in school quality than changes in any other community characteristic [14–16]. If neighborhood schools were disrupted, housing values could decrease by 9.9%, all else remaining constant [12]. Typically, quality schools had high capitalization rates in housing values because of their beneficial relation to children's higher-education prospects [17]. In the rental sector, Beracha and Hardin (2018) believed that rent premium associated with school quality was also statistically significant, merely lower than the premium of housing prices [1].

Besides, school quality capitalization presents spatial heterogeneity. Brasington (2001) emphasized that a school's spatial accessibility can affect its capitalization rate [14]. This capitalization rate is weaker in large communities. Chin and Foong (2006) further argued that accessibility to prestigious schools was a significant determinant of property values [18]. The housing of good accessibility indicates its connection via roads and public transport. Housing prices, in turn, could reflect such conveniences because of shortening commutes. Wen et al. (2018) and Yuan et al. (2020) responded to these previous arguments in the cases of China. They contended that accessibility to schools could affect schools' capitalization in housing prices with a significantly negative correlation [19,20].

### 2.2. Measurements of School Quality Capitalization and Treatment Effects Analyses

Previous literature has discussed many methods to identify and measure school quality capitalization [12], such as the hedonic model [13,14,21,22], the fixed boundary effect method [5], the instrumental method and the differences-in-differences (DID) method [13,23,24]. Feng and Lu (2013) emphasized the importance of considering the endogeneity of school quality when discussing the capitalization of education. Without such consideration, the estimation of school quality capitalization in housing prices might have an upward bias [4].

Many studies introduced the propensity score matching (PSM) method to solve model misspecification and reduce the bias in estimation of the neighborhood effects [25–27]. For example, Wodtke et al. (2011) estimated time-varying treatment effects of exposure to different neighborhoods on high school graduation rates using the inverse probability of treatment weight analysis (IPW), a PSM-based estimator [27]. They found that long-term exposure to disadvantaged neighborhoods during childhood harmed high school graduation rates. Additionally, the IPW estimator performed better in controlling endogeneity problems than a simple linear regression model. Enström Öst and Wilhelmsson (2019) identified different housing situations (e.g., youth apartments, rental apartments) that can cause different fertility and educational patterns by using a propensity score method [28]. Wilhelmsson (2019) estimated the capitalization of energy performance certificates in housing values in Sweden by using nearest neighbor matching (NNM) and radius matching estimators [29]. Zou et al. (2020) employed the PSM method to control migrants' self-selection into different neighborhood types and detected that neighborhood types could impact migrants' socioeconomic integration in Chinese cities. In all, these studies enlighten the applicability of the PSM method to the identification of school quality capitalization in rents [30].

### 2.3. "Tenant Discrimination" in the Housing Market of China

In China, access to neighborhood primary schools is usually determined by the locality of household registration (namely *hukou*) and homeownership within the school district [7]. The *hukou* system is designed in 1958 with the purpose of controlling rural-urban migration and promoting the urbanization process. It indicates two categorical properties of households: locality (local or non-local) and types (rural or urban). Hukou's locality influences households' homeownership and local social benefits, e.g., public education, healthcare, social security [31]. Owner-occupied housing address is linked with the corresponding school district and school enrollments [5,6,32]. In some megacities such as Beijing and Shanghai, where control local *hukou* attainments for migrants [33], purchasing owner-occupied housing is highly affected by household income and having a local *hukou*. Thus, the *hukou* system institutionally contributes to housing inequality between local and non-local *hukou* residents [34–36]. However, the precondition of ensuring the chance of enrollment is to purchase housing with enrollment qualification to the quality school. Without homeownership within the school district, kids from rental households are ranked after those from households of owner-occupied housing in public schools' enrollment. As the number of applicants for quality schools is always exceeding schools' enrollment capacity, rental households are indeed excluded from public education [4]. Thus, this unequal treatment for tenants is so-called "tenant discrimination" in China's housing market [5,6].

### 2.4. School Quality Capitalization in China

Many studies on the capitalization of schools mainly focused on its implicit value in housing prices and explored the positive correlation between school accessibility and housing values in China [4,13,22,23,37–39]. However, only a few studies paid attention to school capitalization in rents. For instance, Zheng et al. (2016) employed the hedonic model to estimate the capitalization of key primary schools in housing prices and rents by using 113 paired data points in Beijing, China [5]. Given the higher density of Beijing's urban structure, they utilized the boundary fixed-effect method to set a maximum distance

of 750 m between paired samples within or beyond school attendance zones. They found that school capitalization in rent was weaker and not significant compared with significant school capitalization in housing prices.

Further, Zhang and Chen (2018) built a hedonic model using resell and rental datasets in Shanghai [6]. They found that the unequal enrollment rights between homeowners and renters can result in significant rent-yield differences across housing size, locational zones, and time. For instance, housing near quality primary schools produced on average 10–35% lower rental yields than housing near ordinary schools. Medium-high income households might regard the low rental yield of owner-occupied housing as the opportunity cost of enrolling in quality schools. They argued that this rent yield gap due to unequal enrollment rights might worsen social inequality between homeowners and renters and contribute to education and residential segregation. Hu et al. (2020) further discussed the social inequality resulting from education accessibility in Shanghai using a time-series rental housing dataset. They used the hedonic model and variance partitioning approach to detect significant school quality capitalization into rents and found that capitalization varied over time. They also identified that housing rents close to quality schools could increase 13.5% compared to housing nearby ordinary school districts after the proposed equitable housing policy by using a difference-in-difference (DID) model [7].

## 3. The Case of Beijing

### 3.1. Rental Housing Market and the Equitable Housing Policy

In China, the rental housing market developed more slowly than the owner-occupied housing sector, particularly the private rental market [40]. The marketization of Chinese housing ended the welfare-oriented housing system in 1998. At the same time, private owner-occupied housings came into the rental market, and institutional housing agencies, such as *Lianjia Company*, have since emerged. However, the rental sector still accounts for a smaller share in the housing market than the owner-occupied sector, commonly regarded as an institutional imbalance between two sectors [41].

Since 2016, the central government has launched a new round of housing market reforms to regulate and develop the rental housing sector. At the end of 2017, Beijing is selected as one of twelve pilot cities in China to implement an equitable housing policy [42]. The equitable housing policy's essence is to grant rental households equal rights to access public primary schools as homeowners and alleviate social inequalities induced by education accessibility [7]. Beijing has conducted the strictest hukou control policy since 2014 among these twelve pilot cities. For instance, the existing point-based *hukou* system in Beijing requires applicants to pay social insurance for seven consecutive years before qualified to apply for local *hukou*, compared with the requirement in Shanghai of seven years in total, and Guangzhou of one year [43]. Given that the housing inequality impacted by *hukou* control is more typical in Beijing, this project selects the city of Beijing as an empirical case to investigate school capitalization in rents.

### 3.2. Beijing Primary School Enrollment Policies

Most of China's primary schools are public schools sponsored by both central and local governments [6]. By 2020, Beijing has 941 primary schools in 16 municipal districts according to the 2020 annual report of the Beijing Municipal Bureau of Statistics. These primary schools' enrollment policies have been neighborhood-based since 1986. Each primary school serves an "attendance zone" or school district, where households are qualified to send their children to a designated neighborhood primary school [5,13]. However, eligibility is mainly determined by homeownership and the parental *hukou* locality (household registration system) in Beijing [36,44–47].

In Beijing, quality primary schools are mainly aggregated in inner urban areas, such as the *Xicheng, Dongcheng, Haidian,* and *Chaoyang* districts. These districts also aggregate a large number of employments and urban amenities. The enrollment policies of high-ranked quality schools in these districts sometimes require both *hukou* and homeownership

within the school district because schools' limited enrollment capacity cannot satisfy all qualified households. Therefore, quality primary schools set ranked priorities for qualified households. For example, households with homeownership in the school district have enrollment priority compared with tenants without Beijing *hukou* or homeownership. Despite the equal rights-oriented housing policy, rental households still encounter difficulties competing for enrollment in neighborhood education, even quality schools. Medium-high income households might expect to invest in rental housing for potential school enrollments. Consequently, such speculative investment would induce a significant rent premium and crowd out low-income tenants to less-central neighborhoods after the implementation of the equitable housing policy [37].

## 4. Data and Models

### 4.1. Study Area and Data

This article selects nine municipal districts of Beijing, the capital of China as the study area, including *Xicheng, Dongcheng, Chaoyang, Haidian, Shiingshan, Fengtai, Tongzhou, Changping*, and *Daxing* districts. Other districts in Beijing are not considered because they are rural area and lack of considerable rental housing lease records. The dataset of 49,438 rental transactions spanning from 2016 to 2018 is provided by *Lianjia Company*, China's largest housing agency. Lianjia's web platform is viewed as a free market of rental housing transactions. On this platform, tenants can appoint the rental housing visit with the brokers, and private landlords can list their owner-occupied housing. The brokers of *Lianjia Company* play a role of housing agency to serve private landlords and tenants and charge services fee based on the leasing contract. Each transaction includes rent and physical attributes (e.g., housing size, the year of construction, the number of bedrooms, living rooms, kitchens, and bathrooms).

In addition, the 171 quality primary schools in our dataset are from the official websites and other social media. According to the schools' reputation and enrollment rate of key middle schools, these schools are classified as popular-class schools, 2nd-class, and 1st-class quality schools. As shown in Figure 1, the 1st-class and 2nd-class quality schools are located in the inner districts of Beijing, involving *Xicheng, Dongcheng, Chaoyang*, and *Haidian districts*. The popular-class quality schools in our dataset are located in the outer districts of Beijing excluding the above four inner districts. From Baidu Map, we further match each school's longitude and latitude coordinates and the rental community with which rental housing units are associated. With these coordinates, we measure the accessibility attributes of rental housing units by the Euclidean distances (e.g., *D_Center, D_Subway, D_Business* and *D_School*), and match amenities within a 500 m radius neighborhood (e.g., the numbers of supermarkets, banks, and restaurants) from Baidu POI (Point of Interest).

Table 1 presents the descriptive statistics and variable definitions. The monthly rent was 6596.21 RMB on average, varying from 1000 to 30,000 RMB (1 RMB ≈ 0.15 USD). The average rental housing unit is 74 square meters, 21.7 years old, with two bedrooms, two living rooms, two kitchens, and two bathrooms. In addition, each rental housing unit is on average 11.7 km from the city center (Tiananmen Square), 0.86 km to the nearest subway station, 2.16 km to the nearest business center, and 0.77 km to the nearest high-quality primary school. Within a 500 m radius, the average rental housing's amenities include 27 supermarkets, 8 bank services, and 248 restaurants.

To address school capitalization, we utilized the boundary fixed effects approach [12,48] and set a 750 m radius buffer from a high-quality school as did in previous literature in the context of Beijing [5,49]. This buffer is regarded as a proxy of the high-quality school district in this paper. Overall, 25,437 rental housing units are within the school district in the dataset, while 24,001 are outside the school district. *School_Ranking* denotes three categories of school quality, that is, popular-class (23.4%), 2nd-class (58.2%), and 1st-class (18.4%). Within a 750 m radius neighborhood, rental housing can have more than one quality school, as shown in Figure 1, which *School_Density* denotes. Overall, 8264 rental housings have more than one quality school, and 17,173 have one neighborhood quality school.

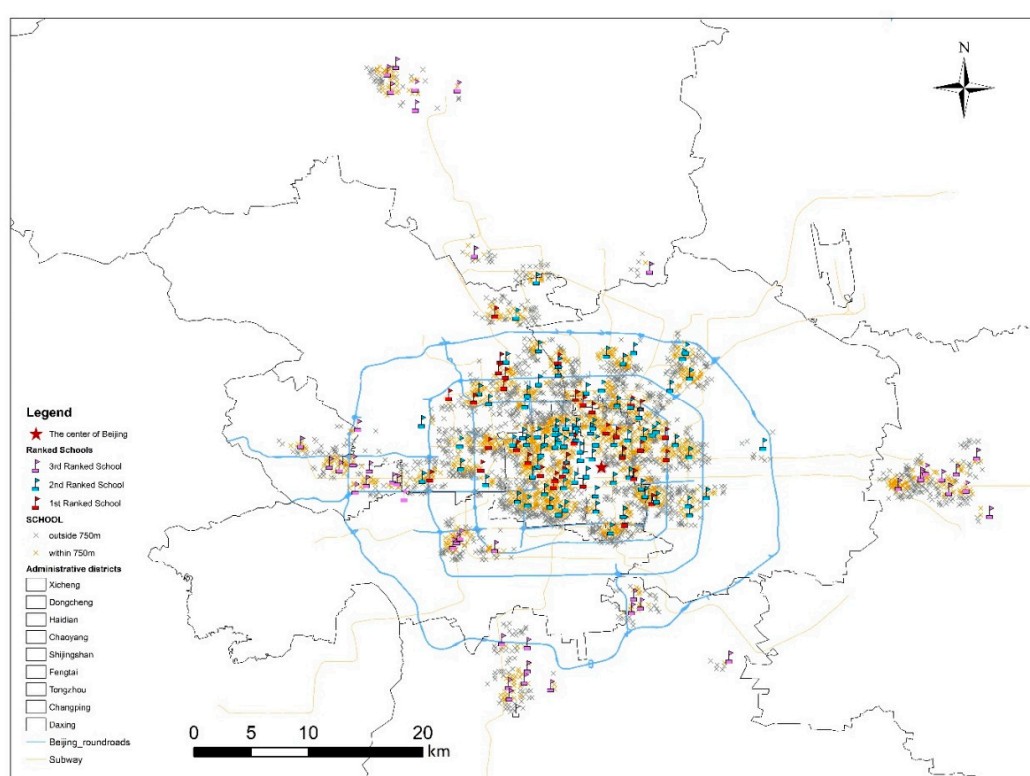

**Figure 1.** Study area and distribution of Beijing rental housing and quality primary schools.

**Table 1.** Descriptive statistics.

| Variable | Definition | Obs | Mean | S.D. | Min | Max |
|---|---|---|---|---|---|---|
| *Rent* | The monthly rent of rental housing unit (RMB/month) | 49,438 | 6596.21 | 3450.23 | 1000 | 30,000 |
| *SCHOOL* | Binary variable, if the nearest quality school is located within 750 m radius neighborhood of rental housing or not | 49,438 | 0.51 | 0.50 | 0 | 1 |
| | Treatment (within 750 m) = 1, if the nearest quality school locates within a 750 m radius neighborhood | 25,437 | 1 | 0 | 1 | 1 |
| | Control (out of 750 m) = 0, otherwise. | 24,001 | 0 | 0 | 0 | 0 |
| *School_Ranking* | Categorical variable, three categories of quality schools' ranking | 49,438 | 1.95 | 0.64 | 1 | 3 |
| | =1, if popular-class schools located in the outer districts; | 11,583 | - | - | - | - |
| | =2, if 2nd-class schools located in the inner districts; | 28,773 | - | - | - | - |
| | =3, if 1st-class schools located in the inner districts; | 9082 | - | - | - | - |
| *School_Density* | Categorical variable, the number of high-quality schools within 750 m neighborhood | 25,437 | 1.32 | 0.47 | 1 | 2 |
| | =1, if there is only one high-quality school | 17,173 | 1 | 0 | 1 | 1 |
| | =2, if there are more than two high-quality schools (two also included) | 8264 | 2 | 0 | 2 | 2 |
| *Size* | The construction area of the rental housing unit (m$^2$) | 49,438 | 73.93 | 31.55 | 9 | 446.38 |
| *Age* | Housing age up to 2019 | 49,438 | 21.72 | 9.79 | 1 | 79 |
| *Bedroom* | The number of bedrooms in the rental housing unit | 49,438 | 1.81 | 0.69 | 1 | 4 |
| *Livingroom* | The number of living rooms in the rental housing unit | 49,438 | 2.02 | 0.43 | 1 | 4 |
| *Kitchen* | The number of kitchens in the rental housing unit | 49,438 | 1.99 | 0.09 | 1 | 3 |
| *Bathroom* | The number of bathrooms in the rental housing unit | 49,438 | 2.11 | 0.34 | 1 | 4 |
| *Supermarket* | Number of supermarkets within a 500 m radius neighborhood | 49,438 | 27.06 | 15.80 | 0 | 158 |
| *Bank* | Number of bank services within a 500 m radius neighborhood | 49,438 | 8.18 | 7.61 | 0 | 118 |
| *Restaurant* | Number of restaurants within a 500 m radius neighborhood | 49,438 | 247.72 | 168.75 | 0 | 2108 |
| *District* | The district where the rental housing unit locates at | 49,438 | 5.13 | 2.62 | 1 | 9 |
| *D_School* | The distance to the nearest high-quality school (km) | 49,438 | 0.77 | 0.36 | 0.01 | 1.50 |
| *D_Center* | The distance to the city center (km) | 49,438 | 11.72 | 7.55 | 0.11 | 40.42 |
| *D_Subway* | The distance to the nearest subway station (km) | 49,438 | 0.86 | 0.76 | 0.1 | 40.17 |
| *D_Business* | The distance to the nearest business center (km) | 49,438 | 2.16 | 2.58 | 0.05 | 55.90 |

*4.2. The Hedonic Rent Model*

Many papers use the hedonic model to measure school capitalization in housing prices [5,14,50]. In the hedonic equation, housing rents can also be decomposed into

different housing characteristics, such as physical housing attributes and neighborhood characteristics [51,52], as shown in Equation (1).

$$\ln Rent_i = \beta_0 + \beta_1 \cdot X_{i1} + \beta_2 \cdot X_{i2} + \beta_3 \cdot X_{i3} + \beta_4 \cdot TD_{i4} + \beta_5 \cdot SCHOOL_i + \varepsilon_i \quad (1)$$

where the dependent variable is the natural logarithm of $Rent_i$ (housing $i$'s rent) based on the transaction when the contract was signed. The independent variables are $X_{i1}$, $X_{i2}$, $X_{i3}$ and $TD_{4i}$. $X_{i1}$ is a vector of physical attributes of rental housing, including the housing size, age, number of bedrooms, living rooms, kitchens, and bathrooms. $X_{i2}$ denotes a vector of the natural logarithm of the distance to the city center, subway, and commercial center. $X_{i3}$ is a vector of amenity attributes, including *supermarkets*, *banks*, and *restaurants*. $TD_{4i}$ are the time dummy variables from February 2016 to April 2018. $SCHOOL_i$ is a binary treatment variable of the quality school. $\beta_1 - \beta_5$ are coefficients to be estimated. $\beta_5$ indicates the treatment effect of school quality, which is its implicit capitalization in rents. $\varepsilon_i$ is the independent and identically distributed error term.

### 4.3. The Propensity Score Matching (PSM) Method

If the relationship between rent and covariates is non-linear, the model structure as shown in Equation (1) would be biased to estimate the capitalization of quality schools. To reduce the bias in estimation and address the endogeneity induced by the model's functional form misspecification (FFM) [29,53,54], this paper introduces the propensity score matching (PSM) method and constructs a hedonic treatment effects model to estimate the capitalization of quality schools.

The PSM method was initiated by Rosenbaum and Rubin in 1983 [55] and is widely used to analyze causal effects with observable nonrandomized data. The propensity score is the probability of treatment assignment on the condition of similar observable covariates between treated and control groups. Using PSM methods requires several preconditions. First, the conditional independence assumption (CIA) [55,56] indicates that, based on the selected covariates, the outcomes from treated and controlled are independent of the treatment. The second precondition is matching or overlapping assumptions [57,58]. Here, the same vector of covariates must identify observations in the treated and controlled groups. The overlapping part of the two groups is called "common support", which predicts the matching estimators' effectiveness to control the omitted variables in different groups [56,59]. That is, sufficient common support is required to achieve an unbiased estimation.

Normally, three types of causal effects are investigated in PSM, such as the average treatment effect (ATE), the average treatment effects in the treated (ATT or ATET), and the average treatment effects on the untreated (ATU or ATC). Here, this project mainly discusses the ATT of quality schools, which can directly reflect the effects of quality schools on rents. Generally, the equation of the propensity score of ATT was written in expectation form, as shown in Equation (2). In this equation, $D$ is the binary treatment (equal to 1 if the rental housing is located in the quality school district; 0 otherwise). $Y^1$ is the outcome value if observable covariates $\delta$ under the treatment $D = 1$; otherwise, $Y^0$ is the outcome if $D = 0$ under the same covariates. That is, $Y^1$ is equal to the rent of rental housing within a quality school district and $Y^0$ is the rent out of a quality school district. Both $Y^1$ and $Y^0$ are under the similarly observable rental housing characteristics.

$$Propensity\_Score_{ATT} = E[\delta | D = 1] = E\left[Y^1 \middle| D = 1\right] - E\left[Y^0 \middle| D = 1\right] \quad (2)$$

PSM provides two measurements to estimate the ATT: using matching and weighting techniques with different estimators. Both matching and weighing estimation rely on a logit or probit regression [55,60]. Matching is a direct way of using the propensity score to discover treatment effects, such as *K*-dimensional nearest neighborhood matching (NNM) [61]. Inverse probability of treatment weighting (IPW) is another estimator generated from the propensity score [62–64]. However, there is no consistent proposal to confirm a PSM estimator for all cases. Thus, the PSM method needs to rest on the research

question and observable covariates. This paper conducts both matching and weighting estimation to check the robustness of the baseline model and identifies the matched sample for heterogeneity analysis of quality schools' capitalization in rents.

## 5. Results

This section first estimates the baseline hedonic rent model using robust standard error. Second, the propensity score method is used to construct the treatment effect hedonic model and investigates the different ranked schools' capitalization in rents. Third, it proceeds by discussing the moderating effects of school density on the nearest school's capitalization. Finally, it identifies the space–temporal heterogeneity of school capitalization in rents by considering different locational zones and periods before and after implementing the equitable housing policy.

### 5.1. The Baseline Regression Results

Table 2 Column (1) reports the baseline result of hedonic regression. The goodness-of-fit is 0.83 (adjusted $R^2$), which indicates an excellent explanatory power, with 83% of dependent variables explained by all the independent variables. The average implicit capitalization of quality schools in rents is 0.0062 at a 1% significance level. It means that the rent of rental housing within a high-quality school district would be increased by 0.62% compared with the rent outside the school district by controlling other covariates equally. This result positively echoes previous literature finding that school quality can significantly impact housing rent [7], but the marginal effects may be smaller than those acting on housing price [1,5]. The low VIF (variance inflation factor) value concerning the binary *SCHOOL* variable indicates a modest multicollinearity problem. Other covariates controlled in the baseline model significantly influence the rent at a 1% significance level. A 1% additional increase in rental housing size can increase the rent by 0.6%, consistent with the statistically significant results from Zhang and Chen (2018) [6]. Further, rental housing's distance to the city center and the nearest subway station are two influential factors in rent. One additional 0.01 km of *D_Center* and *D_Subway* can increase the rent by 0.23% and 0.07%, respectively, reflecting the spatial dependency of housing as discussed in previous studies [6,65].

**Table 2.** The baseline hedonic model and robustness check by propensity score matching and weighting.

| VARIABLE | (1) Default lnR | (2) PSCORE lnR | (3) IPW lnR | (4) Matched lnR |
|---|---|---|---|---|
| *SCHOOL* | 0.0062 *** | 0.0058 *** | 0.0095 *** | 0.0060 *** |
| | (3.75) | (3.53) | (5.20) | (3.61) |
| *pscore* (*SCHOOL*) | - | 0.6322 *** | - | - |
| | | (9.31) | | |
| *lnSize* | 0.5947 *** | 0.6392 *** | 0.5956 *** | 0.5927 *** |
| | (89.64) | (76.35) | (63.27) | (89.39) |
| *lnAge* | −0.1381 *** | −0.1391 *** | −0.1293 *** | −0.1386 *** |
| | (−49.13) | (−49.57) | (−41.42) | (−49.30) |
| *Bedroom* | 0.0510 *** | 0.0266 *** | 0.0551 *** | 0.0519 *** |
| | (20.58) | (7.19) | (16.12) | (20.99) |
| *Livingroom* | 0.0149 *** | 0.0245 *** | 0.0120 *** | 0.0154 *** |
| | (5.61) | (8.69) | (4.23) | (5.84) |
| *Kitchen* | −0.1382 *** | −0.0781 *** | −0.1256 *** | −0.1431 *** |
| | (−6.28) | (−3.38) | (−5.28) | (−6.51) |
| *Bathroom* | 0.1338 *** | 0.1468 *** | 0.1380 *** | 0.1331 *** |
| | (33.06) | (34.32) | (24.45) | (32.92) |
| *lnD_center* | −0.2294 *** | −0.2000 *** | −0.2335 *** | −0.2281 *** |
| | (−60.41) | (−40.62) | (−51.17) | (−60.08) |

**Table 2.** *Cont.*

| VARIABLE | (1) Default lnR | (2) PSCORE lnR | (3) IPW lnR | (4) Matched lnR |
|---|---|---|---|---|
| *lnD_subway* | −0.0658 *** | −0.0050 | −0.0603 *** | −0.0649 *** |
| | (−35.57) | (−0.74) | (−28.65) | (−35.12) |
| *lnD_business* | −0.0275 *** | −0.0481 *** | −0.0302 *** | −0.0265 *** |
| | (−20.20) | (−18.40) | (−18.19) | (−19.54) |
| *Supermarket* | −0.0026 *** | −0.0047 *** | −0.0026 *** | −0.0026 *** |
| | (−31.14) | (−18.93) | (−20.31) | (−31.49) |
| *Bank* | 0.0035 *** | −0.0004 | 0.0040 *** | 0.0043 *** |
| | (20.04) | (−0.95) | (21.66) | (24.98) |
| *Restaurant* | 0.0001 *** | 0.0003 *** | 0.0001 *** | 0.0001 *** |
| | (17.28) | (15.26) | (12.38) | (16.39) |
| *Constant* | 6.6052 *** | 5.9795 *** | 6.5570 *** | 6.6164 *** |
| | (129.89) | (69.25) | (107.61) | (130.14) |
| Fixed district effects | Yes | Yes | Yes | Yes |
| Fixed time effects | Yes | Yes | Yes | Yes |
| Control propensity score | No | Yes | No | No |
| Control inverse probability weight | No | No | Yes | No |
| VIF (*SCHOOL*) | 1.1 | 1.1 | 1.01 | 1.1 |
| No. of observations | 49,438 | 49,438 | 49,438 | 49,324 |
| R-sq | 0.830 | 0.830 | 0.823 | 0.831 |
| adj. R-sq | 0.830 | 0.830 | 0.822 | 0.831 |

Note: $t$ statistics in parentheses: * $p < 0.10$, ** $p < 0.05$, *** $p < 0.01$.

## 5.2. Model Misspecification Adjustment

In conducting the PSM method, the quality school is regarded as a treatment for rent. The propensity scores with a replacement matching approach are calculated for rental housing in the school district (the treated group) and outside the school district (the control group). Figure 2a displays the normality distribution of propensity score under the one-to-one nearest neighbor matching. Figure 2b displays the matching results between treated and controlled groups, including some outliers that cannot be matched in the treated group. The covariates used in the propensity score method include all independent variables in the hedonic model except for *SCHOOL*. The municipal districts and time dummy variables are also directly included.

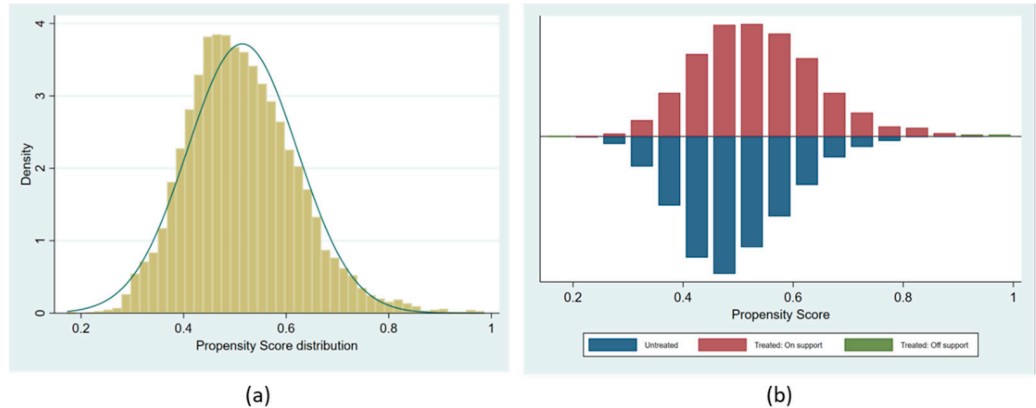

|     |     |
|:---:|:---:|
| (a) | (b) |

**Figure 2.** Propensity score distribution and balancing property. Subfigure (**a**) denotes the normal distribution of the propensity score; subfigure (**b**) shows the matching results between treated and controlled groups.

Figure 3 shows the balancing properties of covariates, which were quite balanced with smaller bias after propensity score matching, compared with the raw (unmatched) ones. A more specific comparison of the mean and variance ratio between treated and

controlled groups is reported in Table 3. The selection bias of the distance to the city center is significantly reduced by 79%. The minimum reduction of the absolute bias of means is 27.3% among all covariates except for the number of bedrooms. The variance ratio between treated and controlled groups is approximately 0.67–1.31. Here, the covariates used in the propensity score method are just the same forms as those in the hedonic model. To achieve the optimal balancing of covariates, one must pre-test the numbers and forms of covariates, such as their linear and quadratic terms [66]. As our pre-testing results, there is no significant differences by choosing simple or quadratic forms of these covariates in estimation of school capitalization in rents. Thus, the paper uses the simple forms of covariates to estimate the propensity score.

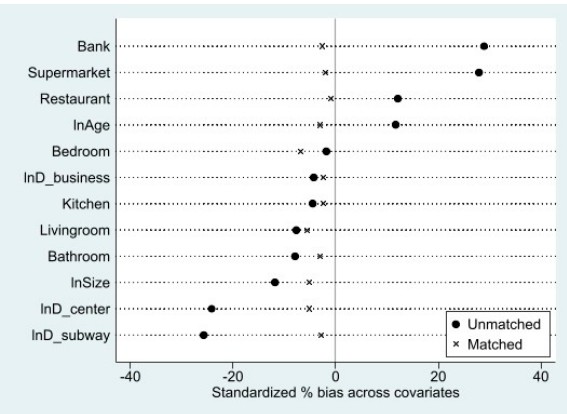

**Figure 3.** Covariates balancing before and after using propensity score matching.

**Table 3.** Balancing property of unmatched and matched covariates using the nearest neighbor PSM.

| VARIABLE | Unmatched (U)/ Matched (M) | Mean Treated | Control | %Bias | % Reduced \|Bias\| | *t*-Test | V(T)/V(C) |
|---|---|---|---|---|---|---|---|
| *lnSize* | U | 4.21 | 4.25 | −11.8 | | −13.1 | 1.01 |
| | M | 4.21 | 4.23 | −5.1 | 56.9 | −5.72 | 1.02 |
| *lnAge* | U | 3.00 | 2.95 | 11.7 | | 12.99 | 0.96 * |
| | M | 3.00 | 3.01 | −3 | 74.1 | −3.42 | 0.98 |
| *Bedroom* | U | 1.80 | 1.82 | −1.9 | | −2.07 | 0.99 |
| | M | 1.80 | 1.85 | −6.9 | −271.3 | −7.77 | 0.99 |
| *Livingroom* | U | 2.01 | 2.04 | −7.7 | | −8.53 | 0.96 * |
| | M | 2.01 | 2.03 | −5.6 | 27.3 | −6.42 | 1.06 * |
| *Kitchen* | U | 1.99 | 2.00 | −4.5 | | −4.94 | 1.69 * |
| | M | 1.99 | 2.00 | −2.4 | 45.8 | −2.6 | 1.31 |
| *Bathroom* | U | 2.10 | 2.13 | −8 | | −8.85 | 0.81 * |
| | M | 2.10 | 2.11 | −3.1 | 61.6 | −3.51 | 0.88 * |
| *lnD_center* | U | 2.21 | 2.35 | −24.2 | | −26.85 | 1.08 * |
| | M | 2.21 | 2.24 | −5.1 | 79.1 | −5.63 | 1.02 |
| *lnD_subway* | U | −0.36 | −0.23 | −25.7 | | −28.59 | 0.76 * |
| | M | −0.36 | −0.34 | −2.8 | 89.2 | −3.16 | 0.80 * |
| *lnD_business* | U | 0.39 | 0.42 | −4.2 | | −4.72 | 0.68 * |
| | M | 0.39 | 0.41 | −2.4 | 42.7 | −2.72 | 0.67 * |
| *Supermarket* | U | 29.18 | 24.82 | 28 | | 31 | 1.39 * |
| | M | 28.93 | 29.24 | −2 | 92.8 | −2.04 | 0.82 * |
| *Bank* | U | 9.23 | 7.07 | 28.9 | | 31.96 | 2.12 * |
| | M | 8.93 | 9.12 | −2.6 | 91.2 | −2.95 | 0.98 |
| *Restaurant* | U | 257.64 | 237.22 | 12.1 | | 13.47 | 0.95 * |
| | M | 256.37 | 257.90 | −0.9 | 92.5 | −1.02 | 0.92 * |

Note: *t* statistics in parentheses: * *p* < 0.10, ** *p* < 0.05, *** *p* < 0.01. \|bias\| means the absolute of bias. V(T)/V(C) denotes the ratio between the variance of the treated group and the variance of the control group.

Next, this paper uses one-to-one-nearest neighbor matching and inverse probability weight (IPW) directly in the hedonic model to adjust model misspecification. In addition, it also estimates the baseline hedonic model using the matched samples. Table 2 *Columns (2)–(4)* exhibit these three models' results. The goodness-of-fit of these models is

0.823–0.831 percent, which indicates the models' good explanatory power. The average capitalization of school quality is approximately 0.0058–0.0095 at a 1% significant level, which reflects the robustness of the baseline hedonic model. Model (4) directly removes 114 outlier samples compared with the baseline model, but the model's adjusted $R^2$ is increased by 0.12%. The VIF value of *SCHOOL* treatment is the same 1.1 as it is in Model (1). Relying on the samples matched through the PSM method, this article proceeds with the heterogeneity analysis of different quality schools relying on Model (4) in subsequent subsections.

*5.3. Heterogeneity Analysis of School Quality Capitalization*

5.3.1. Differences in Ranked Schools' Capitalization

This subsection detects whether different ranked quality schools would be diversely capitalized in rents. It estimates the hedonic treatment-effect model using the matched samples and controlling the urban district effects, time effects, and other neighborhood characteristics to identify schools' capitalization across quality rankings. Table 4 displays the results. The goodness-of-fit (adjusted $R^2$) is approximately 0.798–0.802 percent. All estimations of school quality capitalization are significant at a 1% significance level. For rental housing located in popular-class school districts, rent increases by 3.21% compared with those outside this school district. One acceptable reason for this result would be that popular-class schools are all located in the outer municipal districts of Beijing, where education resources are less concentrated. However, schools in the inner municipal districts performed differently in capitalization in rents. For instance, 2nd-class quality schools decrease rents by 0.75% compared to rent outside the school district. Inversely, the 1st-class schools' capitalization can increase rents within this school district by 2.98%. That may be because the 1st-class quality schools were quite competitive for homeowners and renters due to their top reputation in the inner municipal areas of Beijing. In contrast, 2nd-class quality schools might be not as competitive as 1st-class schools.

**Table 4.** Ranked schools' capitalization in matched treatment effect model.

| VARIABLE | (1) Matched lnR | (2) Popular-Class School lnR | (3) 2nd-Class School lnR | (4) 1st-Class School lnR |
|---|---|---|---|---|
| *SCHOOL* | 0.0060 *** | 0.0321 *** | −0.0075 *** | 0.0298 *** |
| | (3.61) | (11.52) | (−3.39) | (7.94) |
| *lnSize* | 0.5927 *** | 0.4022 *** | 0.6084 *** | 0.7091 *** |
| | (89.39) | (35.71) | (68.14) | (45.45) |
| *lnAge* | −0.1386 *** | −0.1850 *** | −0.1373 *** | −0.1175 *** |
| | (−49.30) | (−42.03) | (−35.21) | (−16.16) |
| *Bedroom* | 0.0519 *** | 0.0755 *** | 0.0509 *** | 0.0312 *** |
| | (20.99) | (18.19) | (15.30) | (5.36) |
| *Livingroom* | 0.0154 *** | 0.0176 *** | 0.0247 *** | 0.0014 |
| | (5.84) | (4.39) | (6.79) | (0.20) |
| *Kitchen* | −0.1431 *** | −0.1652 *** | −0.1364 *** | −0.1631 ** |
| | (−6.51) | (−3.30) | (−5.45) | (−2.48) |
| *Bathroom* | 0.1331 *** | 0.1299 *** | 0.1363 *** | 0.1079 *** |
| | (32.92) | (17.46) | (26.37) | (10.55) |
| *Constant* | 6.6164 *** | 8.2305 *** | 6.6583 *** | 6.4300 *** |
| | (130.14) | (67.23) | (113.52) | (41.79) |
| Fixed urban district effects | Yes | Yes | Yes | Yes |
| Fixed time effects | Yes | Yes | Yes | Yes |
| Control the accessibility attributes | Yes | Yes | Yes | Yes |
| Control the amenity attributes | Yes | Yes | Yes | Yes |
| No. of observations | 49,324 | 11,582 | 28,721 | 9021 |
| R-sq | 0.831 | 0.799 | 0.789 | 0.803 |
| adj. R-sq | 0.831 | 0.798 | 0.788 | 0.802 |

Note: *t* statistics in parentheses: * $p < 0.10$, ** $p < 0.05$, *** $p < 0.01$.

5.3.2. Moderating Effects of Quality School Density on Capitalization

As shown in Figure 1, the location of more than one quality school within rental housing neighborhoods might impact the nearest high-quality school's capitalization in rents. Thus, two interaction terms between ranked schools and school density are introduced into the hedonic model. Table 5 reported the results of school density's moderating effects on school capitalization. The goodness-of-fit, 0.826, indicated an excellent explanation by the

model. School density can noticeably increase the rent by 3.2 percent at a 1% significance level. For those schools within rental housing neighborhoods, the 1st-class quality schools can significantly increase rent by 3.2%, all else being constant, compared with popular-class quality schools. However, 2nd-class schools might not have a significant impact on rent. In addition, the interaction terms revealed the moderating effects of school density, which showed a negative moderation with the increase of school ranking at a minimum 5% significance level.

**Table 5.** Moderating effects of school density on school quality capitalization.

| VARIABLE | Moderating Effects Model lnR |
|---|---|
| *2nd-class School (Popular-class School is the default)* | −0.0031 |
| | (−0.31) |
| *1st-class School* | 0.0823 *** |
| | (8.04) |
| *School_Density (≥2; school number =1 is default)* | 0.0320 *** |
| | (6.17) |
| *Interaction (2nd-class School#School_Density)* | −0.0147 ** |
| | (−2.40) |
| *Interaction (1st-class School#School_Density)* | −0.0612 *** |
| | (−7.90) |
| *lnSize* | 0.5958 *** |
| | (63.94) |
| *lnAge* | −0.1416 *** |
| | (−35.62) |
| *Bedroom* | 0.0641 *** |
| | (18.89) |
| *Livingroom* | 0.0185 *** |
| | (5.00) |
| *Kitchen* | −0.1601 *** |
| | (−5.72) |
| *Bathroom* | 0.1209 *** |
| | (20.72) |
| *Constant* | 6.7815 *** |
| | (102.79) |
| Fixed urban district effects | Yes |
| Fixed time effects | Yes |
| Control the accessibility attributes | Yes |
| Control the amenity attributes | Yes |
| No. of observations | 25,323 |
| R-sq | 0.827 |
| adj. R-sq | 0.826 |

Note: $t$ statistics in parentheses: * $p < 0.10$, ** $p < 0.05$, *** $p < 0.01$.

As shown in Figure 4, neighborhood school density has a positive moderating effect on the popular-class quality schools, increasing school capitalization by 3.5% on average. Still, its moderating effects are weaker on the 2nd-class schools, increasing school capitalization by around 1.5%. This moderating effect performs inversely for the 1st-class schools and can decrease almost 3% of school capitalization. It would be reasonable that the 1st-class schools with the best reputations are located in the inner districts where residents and rental housing are also aggregated.

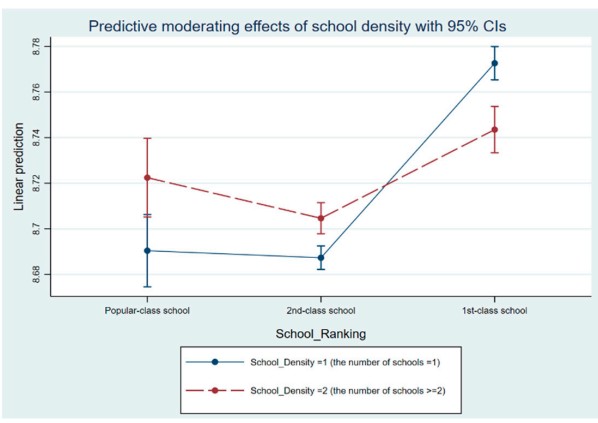

**Figure 4.** School density's moderating effects on school ranking.

### 5.3.3. School Capitalization in Different Segmented Zones

This paper further investigates how school capitalization varies across different residential zones divided by the district boundaries. One motivation of this investigation is that school districts serve households who either register *hukou* or have owner-occupied housing within this school district. However, the district boundary can distinguish the locality of household *hukou*. Thus, quality schools nearby district boundaries might perform different capitalization in rents. As shown in Figure 5, six residential zones are recognized according to the spatial distribution of quality schools and rental housing. *Zones 1–3* aim to identify schools' capitalization within the inner or outer district boundary, and *Zones 4–6* are three cross-boundary areas aiming to detect schools' capitalization across the district boundary. This paper uses the PSM-matched samples to estimate a zone-segmented hedonic model. As displayed in Table 6, the goodness-of-fit of all models is approximately 0.78–0.86. In *Zones 1–3*, the estimations of school capitalization in rents are consistent with Table 4 at a 1% significance level.

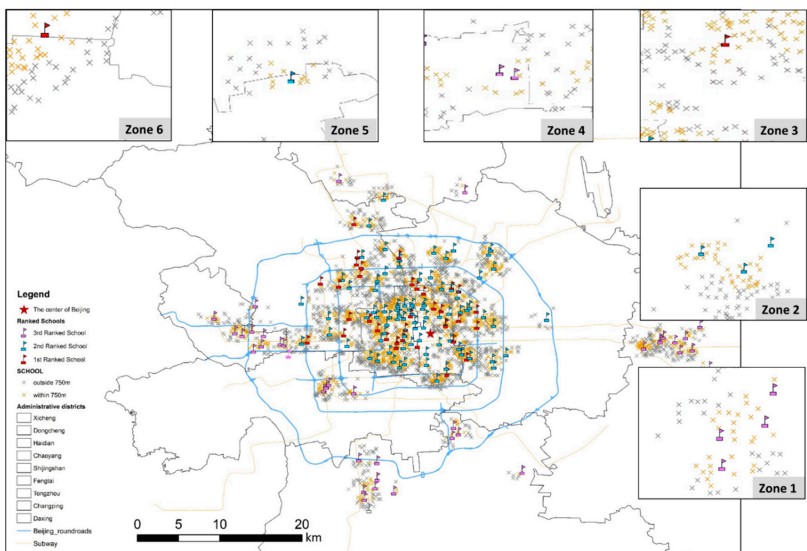

**Figure 5.** Beijing municipality and rental housing (RH)—quality school (QS) zones. Note: *Zone 1* includes the outer municipal area where the popular-class quality schools and rental housings are located; *Zone 2* is the inner municipal area where the 2nd-class schools and rental housings are located; *Zone 3* denotes the municipal area where the 1st-class schools and rental housing are located; *Zone 4* denotes the cross-boundary area of inner districts where the popular-class schools and rental housing are located; *Zone 5* denotes the cross-boundary area of outer districts where the 2nd-class schools and rental housing are located; and *Zone 6* denotes the cross-boundary area of outer districts where the 1st-class schools and rental housing are located.

**Table 6.** School capitalization in different spatial zones.

| VARIABLE | Zone 1 lnR | Zone 2 lnR | Zone 3 lnR | Zone 4 lnR | Zone 5 lnR | Zone 6 lnR |
|---|---|---|---|---|---|---|
| *SCHOOL* | 0.0334 *** | −0.0083 *** | 0.0314 *** | 0.1157 *** | 0.0082 | −0.0600 *** |
| | (11.97) | (−3.68) | (8.09) | (3.59) | (0.65) | (−3.07) |
| *lnSize* | 0.4058 *** | 0.6158 *** | 0.7120 *** | 0.4220 *** | 0.3435 *** | 0.4547 *** |
| | (35.73) | (67.59) | (43.73) | (5.74) | (14.60) | (10.46) |
| *lnAge* | −0.1831 *** | −0.1374 *** | −0.1171 *** | −0.1369 * | −0.0723 *** | −0.1751 *** |
| | (−41.89) | (−34.50) | (−15.37) | (−1.94) | (−4.71) | (−6.71) |
| *Bedroom* | 0.0753 *** | 0.0489 *** | 0.0310 *** | 0.0909 * | 0.1084 *** | 0.0728 *** |
| | (18.11) | (14.38) | (5.13) | (1.94) | (10.70) | (4.20) |
| *Livingroom* | 0.0162 *** | 0.0230 *** | −0.0004 | −0.0151 | 0.0472 *** | 0.0360 * |
| | (4.03) | (6.22) | (−0.06) | (−0.29) | (3.24) | (1.81) |
| *Kitchen* | −0.1636 *** | −0.1391 *** | −0.1842 ** | 0.0000 | −0.0386 | 0.0816 |
| | (−3.26) | (−5.46) | (−2.48) | (.) | (−0.44) | (1.33) |
| *Bathroom* | 0.1267 *** | 0.1372 *** | 0.1083 *** | 0.0675 | 0.0828 *** | 0.0521 * |
| | (17.16) | (26.28) | (10.37) | (1.54) | (2.70) | (1.67) |
| *lnD_center* | −0.4521 *** | −0.2441 *** | −0.1782 *** | −1.9685 *** | −0.0441 * | −0.2153 * |
| | (−33.82) | (−51.60) | (−20.25) | (−6.14) | (−1.86) | (−1.74) |
| *lnD_subway* | −0.1128 *** | −0.0420 *** | −0.0643 *** | 0.2419 *** | −0.1066 *** | 0.0186 |
| | (−38.41) | (−15.75) | (−11.70) | (6.79) | (−5.74) | (1.03) |
| *lnD_business* | −0.0028 | −0.0252 *** | −0.0342 *** | 0.1376 ** | −0.0123 | 0.0613 *** |
| | (−0.86) | (−13.99) | (−9.08) | (2.43) | (−1.13) | (3.91) |
| *Supermarket* | −0.0015 *** | −0.0028 *** | −0.0021 *** | −0.0046 | −0.0019 *** | −0.0001 |
| | (−8.50) | (−25.35) | (−9.81) | (−1.23) | (−6.55) | (−0.11) |
| *Bank* | 0.0012 *** | 0.0037 *** | 0.0048 *** | 0.0386 ** | 0.0061 *** | 0.0012 |
| | (2.75) | (15.68) | (14.09) | (2.20) | (3.85) | (0.59) |
| *Restaurant* | 0.0001 *** | 0.0001 *** | 0.0002 *** | 0.0010 | 0.0000 | 0.0002 ** |
| | (7.42) | (10.66) | (10.91) | (1.28) | (0.97) | (1.98) |
| *Constant* | 8.1981 *** | 6.8836 *** | 6.4635 *** | 12.4846 *** | 6.8127 *** | 7.0435 *** |
| | (67.00) | (116.77) | (37.43) | (10.72) | (34.17) | (25.80) |
| Fixed inner district effects | No | Yes | Yes | Yes | No | No |
| Fixed outer district effects | Yes | No | No | No | Yes | Yes |
| Control popular-class schools | Yes | No | No | Yes | No | No |
| Control 2nd-class schools | No | Yes | No | No | Yes | No |
| Control 1st-class schools | No | No | Yes | No | No | Yes |
| Fixed time effects | Yes | Yes | Yes | Yes | Yes | Yes |
| No. of observations | 11,454 | 27,801 | 8535 | 128 | 920 | 486 |
| R-sq | 0.802 | 0.784 | 0.795 | 0.858 | 0.823 | 0.871 |
| adj. R-sq | 0.801 | 0.784 | 0.794 | 0.797 | 0.815 | 0.860 |

Note: *t* statistics in parentheses: * $p < 0.10$, ** $p < 0.05$, *** $p < 0.01$.

However, in *Zones 4–6*, school capitalization shows significant differences. For rental housing units located in the inner districts, the nearest (popular-class) schools are significantly capitalized in rents, which could be increased by 11.57 percent, as shown in *Zone 4*. One possible reason may be that the value of housing located in the inner districts is highly determined by quality schools and other factors, such as the distance to the city center (−1.97 ***) in Table 6. The 2nd-class quality schools do not show significant capitalization in rents of rental housing in the outer districts. The 1st-class quality schools even had a significantly negative impact, decreasing by 6% the rents of nearby rental housing. This result echoes the discussion of "tenant discrimination" that school capitalization is closely connected with homeownership because renters can be excluded from quality schools' enrollment. Additionally, high-ranked quality schools cannot necessarily yield high rents within school districts [6].

### 5.3.4. Ranked Schools' Capitalization before and after the Equitable Housing Policy

School quality capitalization varied not only across space, but also time. We separately estimated the hedonic model by controlling the same period of three months before and after implementing the equitable policy. Table 7 presents the results. Before implementing this policy in November 2017, all schools had a significant capitalization in rents at a minimum 10% significance level. For instance, the capitalization of the popular-class and 2nd-class schools was consistent with our estimation in Table 4, but the 1st-class schools' capitalization was less than in the matched treatment model. However, after the policy, the 1st-class schools' capitalization in rent became higher than the popular-class schools, whose capitalization in rent was weakened, and 2nd-class schools, whose capitalization became not significant. The 1.14 percent increase of the 1st-class schools' capitalization in rent reflected that the equitable right policy might worsen the housing inequality between

homeowners and renters because renters must pay a significant rent premium without obtaining enrollment opportunities [6,7].

**Table 7.** Ranked schools' capitalization before and after the equitable policy.

| VARIABLE | (1) Popular-Class School | | (2) 2nd-Class School | | (3) 1st-Class School | |
|---|---|---|---|---|---|---|
| | lnR Before | lnR After | lnR Before | lnR After | lnR Before | lnR After |
| *SCHOOL* | 0.0315 *** | 0.0289 *** | −0.0088 * | −0.0081 | 0.0159 * | 0.0273 ** |
| | (5.37) | (4.14) | (−1.69) | (−1.26) | (1.75) | (2.56) |
| *lnSize* | 0.4212 *** | 0.4085 *** | 0.6267 *** | 0.5666 *** | 0.6984 *** | 0.6039 *** |
| | (17.28) | (17.84) | (33.78) | (15.30) | (19.21) | (10.76) |
| *lnAge* | −0.1606 *** | −0.1467 *** | −0.1233 *** | −0.1372 *** | −0.1053 *** | −0.1422 *** |
| | (−16.33) | (−14.45) | (−13.65) | (−10.79) | (−6.82) | (−5.52) |
| *Bedroom* | 0.0732 *** | 0.0834 *** | 0.0393 *** | 0.0676 *** | 0.0408 *** | 0.0390 ** |
| | (7.84) | (9.53) | (5.59) | (5.40) | (3.42) | (1.98) |
| *Livingroom* | 0.0072 | 0.0055 | 0.0100 | 0.0163 | −0.0163 | 0.0149 |
| | (0.81) | (0.59) | (1.15) | (1.32) | (−1.03) | (0.60) |
| *Kitchen* | 0.0238 | −0.0642 | −0.0560 | −0.1156 | −0.3370 | −0.5530 ** |
| | (0.36) | (−0.90) | (−1.08) | (−1.11) | (−1.62) | (−2.41) |
| *Bathroom* | 0.1245 *** | 0.1216 *** | 0.1503 *** | 0.1371 *** | 0.0966 *** | 0.1246 *** |
| | (8.57) | (8.04) | (12.77) | (7.56) | (3.74) | (4.41) |
| *lnD_center* | −0.4279 *** | −0.5230 *** | −0.2390 *** | −0.2369 *** | −0.1875 *** | −0.1969 *** |
| | (−15.23) | (−16.12) | (−23.51) | (−14.83) | (−8.75) | (−8.64) |
| *lnD_subway* | −0.1089 *** | −0.0807 *** | −0.0429 *** | −0.0429 *** | −0.0668 *** | −0.0544 *** |
| | (−18.44) | (−11.91) | (−7.02) | (−5.65) | (−5.27) | (−3.13) |
| *lnD_business* | −0.0111 | −0.0035 | −0.0229 *** | −0.0234 *** | −0.0331 *** | −0.0252 ** |
| | (−1.52) | (−0.44) | (−5.04) | (−4.96) | (−3.64) | (−2.28) |
| *Supermarket* | −0.0014 ** | −0.0009 ** | −0.0031 *** | −0.0028 *** | −0.0030 *** | −0.0029 *** |
| | (−2.53) | (−2.01) | (−10.37) | (−9.17) | (−5.75) | (−4.14) |
| *Bank* | −0.0000 | 0.0004 | 0.0033 *** | 0.0030 *** | 0.0044 *** | 0.0054 *** |
| | (−0.04) | (0.33) | (5.30) | (4.17) | (5.58) | (4.23) |
| *Restaurant* | 0.0002 *** | 0.0001 ** | 0.0001 *** | 0.0001 *** | 0.0002 *** | 0.0002 *** |
| | (4.11) | (2.05) | (5.08) | (5.14) | (5.63) | (5.44) |
| *Constant* | 7.7707 *** | 8.2898 *** | 6.5275 *** | 6.8879 *** | 7.0269 *** | 7.8017 *** |
| | (40.89) | (41.19) | (51.05) | (32.58) | (16.71) | (13.32) |
| Fixed urban district effects | Yes | Yes | Yes | Yes | Yes | Yes |
| Fixed time effects | Yes | Yes | Yes | Yes | Yes | Yes |
| No.of observations | 2155 | 1597 | 4825 | 3566 | 1576 | 1128 |
| R-sq | 0.832 | 0.842 | 0.793 | 0.786 | 0.796 | 0.782 |
| adj. R-sq | 0.830 | 0.840 | 0.792 | 0.785 | 0.794 | 0.778 |

Note: *t* statistics in parentheses: * $p < 0.10$, ** $p < 0.05$, *** $p < 0.01$.

## 6. Conclusions

This paper presents comprehensive discussions related to the school quality capitalization in rents using a dataset of 49,438 rental housing transactions from 2016 to 2018 in Beijing, China. Further, it utilizes the propensity score method (PSM) to reduce the model misspecification and construct the hedonic treatment effect models to estimate quality school capitalization. The findings of this paper are consistent with previous studies [5–7,67]. School quality can be significantly capitalized in rents. In addition, it further estimates school capitalization in rents across school quality ranking (1st-class, 2nd-class, and popular-class), spatial zones, and periods before and after implementing the equitable housing policy. The 1st-class and popular-class quality schools are significantly capitalized into rent and promote the rent premium by 2.98 and 3.21 percent, respectively. However, the 2nd-class quality schools show a slightly negative impact because these schools are not as popular as the 1st-class schools in inner municipal districts.

In addition, this article investigates school density's moderating effects on school capitalization within rental housing neighborhoods. Within rental housing neighborhoods, quality school density can significantly moderate the nearest school's capitalization, promoting rents by 3.5% in outer municipal districts but decreasing the 1st-class quality schools' capitalization in rents by 3% in inner districts. To address school capitalization's spatial dependency, it further recognizes six residential zones, finding no significant evidence that 1st-class quality schools are capitalized in the rent of outer municipal areas due to existing tenant discrimination. The popular-class quality schools can also be capitalized into housing rent of inner municipalities, but this might result from other exogenous factors (e.g., housing prices, public transport) as discussed by Zheng et al. (2016) [5] and Zhang et al. (2019) [6]. Finally, we evaluate the variation of school capitalization before

and after the equitable housing policy. In the districts of high competition for 1st-class quality schools, the equitable housing policy shows a possible failure because 1st-class schools' capitalization increased by 1.14% after policy implementation. However, renters still cannot enjoy equal access to quality schools. This result indicates social inequality between homeowners and renters would be worsened [6,7]. In less competitive municipalities for the 2nd-class and popular-class quality schools, the equitable policy seems to be an effective remedy to reduce school capitalization in rents.

Some potential aspects can be addressed in the future based on this paper. For instance, the authors merely use a proxy measurement of rental housing neighborhoods to estimate school quality capitalization due to difficulties accessing data about particular rental housing in particular school districts. Future studies could identify the spatial dependency of school quality capitalization by using spatial lag models (SLM) and spatial error models (SEM). If we can collect more observations of rental housing transactions after the implementation of equitable housing policies, we can further identify school capitalization in rents in the post-policy period. Additionally, it would be of interest to detect school quality capitalization differences between cities under equitable housing policy and those not, assuming panel data is available for future studies.

**Funding:** This research was funded by China Scholarship Council (grant number: CSC201700260251) and National Natural Science Foundation of China (grant number: 72011530136).

**Institutional Review Board Statement:** Not applicable.

**Informed Consent Statement:** Not applicable.

**Data Availability Statement:** The source for the rental transaction data used in this research is from Lianjia Company (https://bj.lianjia.com accessed on 29 March 2022); Data regarding primary schools in Beijing are from the official website (http://www.ysxiao.cn accessed on 29 March 2022) and social media (https://www.sohu.com/a/108411984_256157 accessed on 29 March 2022 and http://beijing.xueda.com/news/197604.shtml accessed on 29 March 2022); Data of coordinates and amenities nearby rental housing is from Baidu Map (https://map.baidu.com/ accessed on 29 March 2022).

**Conflicts of Interest:** The author declares no conflict of interest.

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
