# Peer review of "The Capitalization of School Quality in Rents in the Beijing Housing Market: A Propensity Score Matching Method"

_buildings, doi:10.3390/buildings12040485_

Round 1
Reviewer 1 Report
In the paper the Author undertakes interesting and important especially for big cities like Beijing, topic: The capitalization of school quality in rents in the Beijing housing market. The paper is based on primary data obtain in researches conducted in Beijing in 2016-2018. The findings can be used in property management, management of investments in housing market by developers and in planning housing policy.
However, Author/s should correct the article with minor suggestions listed below:
- Abstract is constructed properly and enclosed: aim of the paper, used methodology, description of researches conducted but in the middle of it is a gap line (between "Further, it confirms that" and "first-ranked schools cannot be significantly capitalized in the rent....") which should be deleted.
- introduction is prepared well and provide the background of the paper
- literature review is wide and adequate
- methodology describe used methods well
- at the end of the paper Results and Discussion are exposed as one chapter 6 I suggest to divide these parts and developed Discussion
- I like that the Author exposed in conclusion the possibility of future studies
- the references should be analyzed and correct, for example position 40
Reviewer 2 Report
The topic of the article is very interesting but the treatment of the research is very complex and not always clear.
- Furthermore, the methodology of the article examines a method that is described as propensity score method (PSM) while the statistical method recognised and developed by Rosembaum and Rubin is defined as Propensity Score Matching.
- There are also parts of the text highlighted in yellow as typos.
- On page 3 it says, "in previous literature", proceeding to what if past references are also given?
- It is not clear to me what is meant by the Brasington (2001) reference. What is meant by the term "the capitalisation of school quality in rents"? As spill-over from the cost of rents? The expression is not clear to me.
- My advice is to revise the structure of the article, making the model and also its explanation clear.
- On page 3 it says "Unlike the homeowner property tax system in European countries and American states, public schools in China are financed instead by both central and local governments", but even in Europe public schools are financed by the central and/or regional government, so I don't understand.
- Hokou, system of registration of things, is not clear how it works.
- Also, in the survey of rental houses it is described that on average flats of an average size of 74m have two kitchens, two living rooms. Why two kitchens, two living rooms? That's strange to me. Is there an explanation?
Reviewer 3 Report
Thank you for giving me the opportunity to review the article. Below my remarks:
- Please edit the title of the article because it is not clear enough: "The capitalization of school quality in rents".
- The authors have chosen an interesting and current topic.
- The abstract and introduction are ok.
- The Literature review chapter is well-crafted and detailed and contains relevant literary sources.
- To ensure objectivity, extend the data file to include data outside Lianjia, China's largest housing agency. No need to get a lot of additional data. It is only a matter of verifying that the data used in the article represent the usual situation on the real estate market.
- In Table 1 the monthly rent was used 6,596.21 RMB on average. I think it would be more appropriate to work with median values.
- In further research, I recommend that separate statistical files will be created for different sizes of apartments (for example 30 m2, 50 m2 and 70 m2). Not just based on an average rental housing unit of 74 square meters.
- Is it really common for an apartment to always have two kitchens and two bathrooms?
Regards,
Reviewer 4 Report
See my referee report attached as a PDF file.

Round 2
Reviewer 2 Report
I have no comments to add.